# Abrasion Behavior of Steel-Fiber-Reinforced Concrete in Hydraulic Structures

**Yu-Wen Liu [1], Yu-Yuan Lin [1] and Shih-Wei Cho [2,*]**

[1]  Department of Civil and Water Resources Engineering, National Chiayi University, Chiayi 600355, Taiwan; yuwen@mail.ncyu.edu.tw (Y.-W.L.); yylin@mail.ncyu.edu.tw (Y.-Y.L.)

[2]  Department of Civil Engineering and Engineering Management, National Quemoy University, Kinmen 892009, Taiwan

*  Correspondence: swcho@nqu.edu.tw; Tel.: +886-82-313300

**Abstract:** This study investigated two types of abrasion resistance of steel–fiber-reinforced concrete in hydraulic structures, friction abrasion and impact abrasion using the ASTM C1138 underwater test and the water-borne sand test, respectively. Three water-to-cementitious-material ratios (0.50, 0.36, and 0.28), two impact angles (45° and 90°), plain concrete, and steel–fiber-reinforced concrete were employed. Test results showed that the abrasive action and principal resistance varied between the two test methods. The average impact abrasion rates (IARs) of concrete were approximately 8–17 times greater than the average friction abrasion rate (FARs). In general, the impact abrasion loss of the concrete surface impacted at a vertical angle was higher than that of impacted at a 45 degree angle. Moreover, the average FAR and IAR decreased when the concrete was reinforced with steel fibers. The steel fibers acted as shields to prevent the concrete material behind the fibers from abrasion, thus improving abrasion resistance. In both the underwater and waterborne sand flow methods, the resistance to abrasion of concrete without steel fibers increased as the water/cementitious material ratio (w/cm) decreased, and the concrete compressive strength also increased.

**Keywords:** abrasion resistance; friction abrasion; impact abrasion; hydraulic structure; steel-fiber-reinforced concrete

## 1. Introduction

As detailed in ACI 210R-93 [1], most of the abrasion erosion harm is attributed to the action of water-borne silt, sand, stone, rock, ice, and other scraps on the surface of concrete during the hydraulic structure operation. In Taiwan, all rivers originate from the peaks of mountain ridges. Due to high altitude mountain and steep valley, all rivers are short and steep, causing rapid flow during storms, especially in the typhoon season. The average annual rainfall in Taiwan is about 2.6 times the world average. Therefore, rainfall is concentrated in the months from May to October, which is about 78% of the average annual rainfall. In addition, due to the frequent earthquakes in Taiwan, the fast-flowing river carries large quantities of sand and gravel, causing the river sediment yield and the percentage of sand in the river to exceed 10 times the global average [2]. Therefore, the main abrasion-erosion harms are caused by friction and the impact of waterborne sand on the surface of hydraulic concrete [3].

Hydraulic concrete structures often sustain surface damage after being subjected to the rapid flow of waterborne sand and gravel over a long period. The masses and flow speeds of these waterborne particles constitute a transient striking momentum, slowly chipping off the concrete surface. This disintegrating behavior is known as abrasive erosion and eventually shortens the service life of hydraulic concrete structures. During field observations (Liu 2005), waterborne sand was one of the main causes of abrasion-erosion damage in hydraulic structures other than large stone impacts.

Figure 1 illustrates the abrasion process of a concrete surface subjected to a high speed impingement jet of waterborne sand. First, the part of the concrete that belongs to the surface of the mortar gradually wears away., subsequently exposing the coarse aggregate. Next, due to the impact of waterborne particles, the broken coarse aggregate results in the formation of tiny voids in the interface between the paste and aggregate. After repeated impacts, voids are formed, which further derive into the concrete. The formation of voids is profoundly affected by the size of the coarse aggregate, the type of sand used, and the momentum of the rotating water jet. The abrasion rate depends on the hardness, shape, size, and quantity of the waterborne particle; the speed of the water jet; and concrete quality. Although good-quality concrete can withstand the impact of high flow velocity of jet and has almost no damage, the concrete cannot withstand the abrasion effect of scraps grinding or repeated impact on the surface for a long time. The fact that water-cement ratio, cement amount in concrete, slump, air content, type of coarse aggregate, type of finish, fly ash, silica fume, fibers, and curing affect the characteristics of a concrete surface layer has long been recognized. Because these factors also influence the compressive strength of concrete, a tendency to accept concrete strength as a criterion for abrasion resistance evaluation is evident in references [4–7]. Such as Pyo et al. [8] discussed the influence of gravels on the abrasion of high strength concrete. Test results shows the high strength concrete with gravels shows a lower abrasion resistance than the high strength concrete with no gravels. Pyo et al. show that the abrasion resistance is influenced by the interface between the aggregate and the paste.

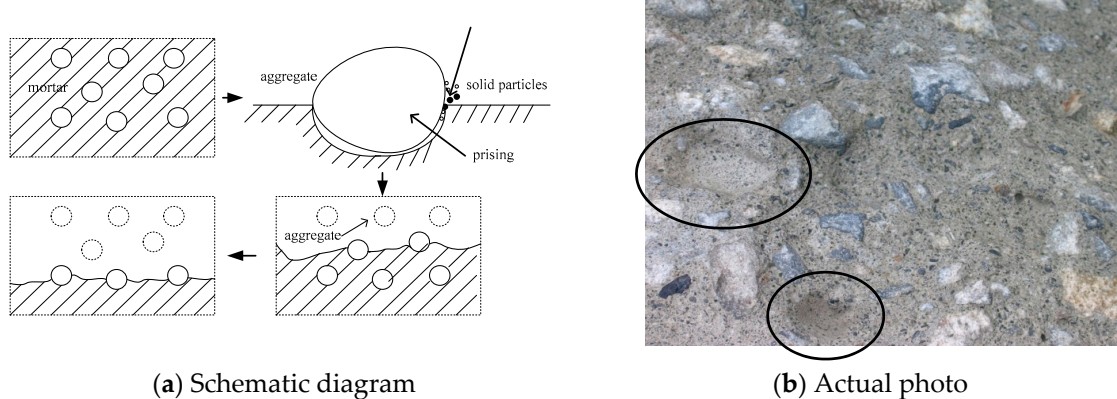

(**a**) Schematic diagram          (**b**) Actual photo

**Figure 1.** Abrasion process and abrasion damage of hydraulic concrete surface.

Since many types of abrasion become concrete damage problem, there are many abrasion test methods. Some researchers [9–13] use current commonly used test methods to determine frictional attrition under specific conditions, including the case of limiting amount of tiny grains on a small surface area of concrete. In general, the abrasion resistance is determined by the interface properties between the paste and aggregate. A water jet containing an appropriate amount of sand can be used to compare the actual abrasion and erosion of concrete to improve the existing abrasion method.

This study explored the friction and impact abrasion behavior of hydraulic concrete made with and without steel fibers, using the standard ASTM C1138 [14] test method (underwater method) and water-borne sand flow impact test method, respectively. Additionally, the results of abrasion loss, (average) friction abrasion rate, (average) impact abrasion rate, and correlation between compressive strength and abrasion properties (average abrasion rate) are presented.

## 2. Materials and Methods

### 2.1. Materials and Mixture Proportion

Materials used in manufacturing test slabs are outlined as follows: (1) Type I Portland cement (ASTM C150), chemical compositions as listed in Table 1; (2) river sand (fineness: 2.95; specific gravity: 2.66; and absorption: 1.2%); (3) gravel (maximum aggregate size (Dmax): 19 mm; specific gravity:

2.64; absorption: 1.0%; and dry-rodded density: 1630 kg/m$^3$); (4) ground-granulated blast furnace slag (GGBFS; specific gravity: 2.89), chemical compositions as listed in Table 1; (5) silica fume (SF, specific gravity: 2.2) meeting the requirements of ASTM C 1240), chemical compositions as listed in Table 1; (6) steel fiber properties, as listed in Table 2; (7) superplasticizer (SP) fitting to ASTM C494 Type-G (specific gravity: 1.1); and (8) fresh water.

**Table 1.** Chemical compositions of cement, GGBFS, silica fume (%).

| Materials | SiO$_2$ | Al$_2$O$_3$ | CaO | Fe$_2$O$_3$ | Na$_2$O | K$_2$O | SO$_3$ | MgO |
|---|---|---|---|---|---|---|---|---|
| Cement | 20.9 | 5.6 | 63.9 | 3.1 | – | – | 2.5 | 2.9 |
| GGBFS | 33.7 | 13.7 | 42.1 | 1.2 | 0.29 | 0.3 | 1.87 | 6.53 |
| Silica fume | 93.2 | 0.1 | 0.9 | 0.04 | 0.6 | 2.0 | 0.005 | 0.5 |

**Table 2.** Properties of steel fibers.

| Fiber Shape | Coating | Diameter, mm | Length, mm | Tensile Strength, MPa | Specific Gravity |
|---|---|---|---|---|---|
| End-hooks | Brass | 0.75 mm | 30 | 1050 | 7.8 |

The mixture proportions used in this investigation are listed in Table 3. For plain concrete, mixtures were prepared with three different water/cementitious material ratios (w/cm) of 0.28, 0.36, and 0.50. The high-strength concrete was simulated with a w/cm ratio of 0.28. The w/cm ratio of 0.36 was used to simulate the common design strength of hydraulic concrete, and the w/cm ratio of 0.5 was normal concrete. For each w/cm ratios, cement was partially replaced with 20% of GGBFS by mass. The GGBFS addition is according experience suggests [15]. For steel fiber concrete, a w/cm ratio of 0.36 was used. A silica fume addition 20% by weight of (cement + GGBFS) and steel fiber addition 1% by total volume were selected for designing the mixture proportions to meet abrasion resistance requirements. The workability of each mixtures of concrete was maintained by the superplasticizer with a slump of approximately 21 ± 3 cm.

**Table 3.** Concrete mixture proportions and results of compressive strength.

| Batch | w/cm | Quantities, kg/m$^3$ | | | | | | | | Slump, cm | Compressive Strength, MPa |
|---|---|---|---|---|---|---|---|---|---|---|---|
| | | Water | Cement | GGBFS | SF | Steel Fiber | Sand | Gravel | SP | | |
| C28 | 0.28 | 160 | 457 | 114 | 0 | 0 | 730 | 925 | 12.5 | 24 | 90.8 |
| C36 | 0.36 | 160 | 356 | 89 | 0 | 0 | 780 | 985 | 10.9 | 22 | 50.3 |
| C50 | 0.50 | 160 | 288 | 32 | 0 | 0 | 840 | 1055 | 0.5 | 21 | 30.4 |
| SC36 | 0.36 | 160 | 285 | 71 | 89 | 78 | 755 | 945 | 15 | 18 | 51.0 |

### 2.2. Casting

For this research, following specimens were cast: (a) Three 15 cm (diameter) × 30 cm (height) cylindrical specimens were made and tested for compressive strength testing in accordance with ASTM C39. (b) Three 30 cm (diameter) × 10 cm (height) disk-shaped specimens were made for friction abrasion tests in accordance with ASTM C1138 (1997). (c) Three square slabs, 20 cm × 20 cm × 50 mm (thickness) for the water-borne sand flow impact test. The measured average abrasion rate of three plates was designated as the representative data for each concrete mixture for reference. Twenty-four hours after the speimens were cast, they were curried in water under 23 ± 3 °C. Tests were performed after 28 days of curing. Table 2 lists the compressive strength of the produced concrete specimens.

### 2.3. ASTM C1138 Test Method

The ASTM C1138 (1997) "Standard Test Method for Abrasion Resistance of Concrete (Underwater Method)" test method is intended to qualitatively simulate the behavior of swirling water containing suspended and transported solid objects that produce abrasion of concrete and cause

potholes and related effects. The wearing action caused by rolling steel balls simulates the friction action of waterborne particles on concrete surfaces, but not the impact action [14].

The apparatus components are outlined as follows: a drill press; an agitation paddle; a cylindrical steel container, measuring 305 ± 6 mm (inside diameter) × 450 ± 25 mm (height), housing a disk-shaped concrete specimen; and 70 steel grinding balls of various size (ten 25-mm-diameter balls, thirty-five 19-mm-diameter balls, and twenty-five 13-mm-diameter balls). The water in the container is circulated by the immersed agitation paddle powered by the drill press rotating at a speed of 1200 ± 100 rpm. The circulating water, in turn, moves the steel grinding balls on the concrete surface, producing the desired abrasion effects.

The underwater friction abrasion test involved six 12 h periods for a total of 72 h. At the end of every 12 h of operation, the specimen was removed from the test container, the abraded material was flushed off, and the surface was dried. This study determined and recorded the specimen mass in air to the nearest 10 g. The abrasion loss (percent by mass), friction abrasion rate (FAR, in g/h), and average FAR was determined from the specimen mass loss versus mass before test, mass loss versus test period, and total mass loss versus test time, respectively. A minimum of three specimens were used to establish the FAR. The range among the test results for the three specimens should be no greater than 46% of their average.

### 2.4. Waterborne Sand Test Method

The waterborne sand test method developed in this program was specially designed to evaluate the abrasion resistance of concrete surfaces subjected to the impact and impingement of water flow that carries a predetermined amount of sand. To accurately model the damage mechanisms, a carefully thought-out test procedure was conducted by combining the impact load of a water jet and the shear/friction forces of sand particles that accompany it.

To understand the interfacial bonding behaviors between coarse aggregate and mortar, a specially designed and fabricated 10 mm × 200 mm rectangular nozzle sufficiently large to cover the maximum aggregate size was used in the waterborne sand tests (Figure 2). A rectangular nozzle was used because it produces a water-jet flow over a spillway in the field as opposed to a circular flow.

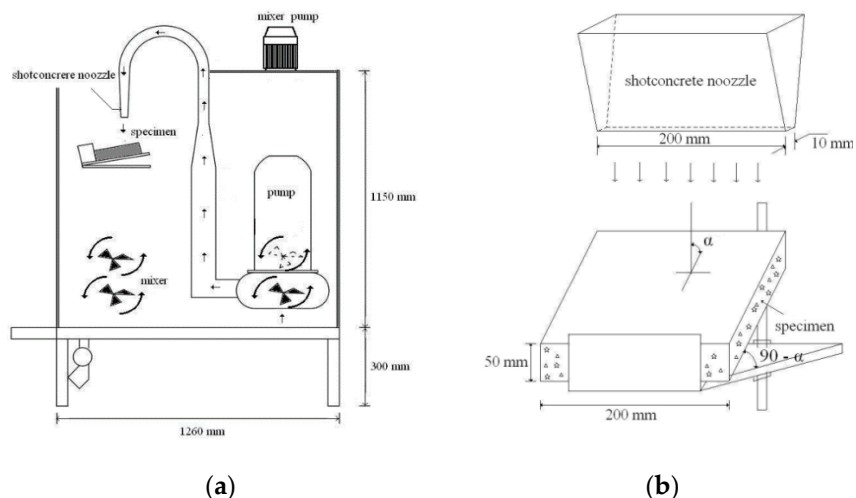

(**a**)　　　　　　　　　　　　　　　　(**b**)

**Figure 2.** (**a**) Test apparatus for waterborne sand flow impact abrasion, (**b**) Schematic diagram of nozzle.

The test water was made by mixing quartz river sand no coarser than 5 mm to formulate a slurry mixture. During the tests, the nozzle was held at 90° and 45° angles in relation to the test slab to evaluate the effects of flow angles on the impact abrasion.

An abrasion chamber measuring 1260 mm × 1150 mm × 1170 mm was installed. The chamber can accommodate four individual pumps that can simultaneously shoot four separate water flows of different sand mixtures at different velocities onto the test slabs that were positioned above water level.

Fresh sand supply was used to form the designed water flows composed of angular quartz-rich river sand with a Mohs-hardness (Hp) of 8 and specific gravity of 2.64. In general, sand was gradually poured in and mixed for 5 min until the mixture reached 300 kg/m$^3$ sand content.

During each water jet test, the cavitation index was first assessed and found to be 0.2. In accordance with reference (ACI 210R-03, 2003), a cavitation index of 0.2 is small enough to be ignored. Throughout the 2 h water jet test, the water velocity was controlled at 10 m/s, which is equivalent to a 0.17-MPa pressure on the test slab, and the water temperature was maintained at 30 °C.

Immediately after the test, the loose materials were flushed and collected to determine their mass with a precision of ±0.05 g. This study also measured the mass of the slab before (m1) and after (m2) the test to determine the impact abrasion loss (percent by mass); moreover, the impact abrasion rate (IAR, in g/h) and average IARs we determined from the specimen's mass loss versus test period and total mass loss versus test time, respectively. A minimum of three measurements were used to establish the IAR. The range among the test results for the three specimens should be no greater than 45% of their average to quickly determine whether the test process is successful.

## 3. Results and Discussion

### 3.1. Difference between Underwater Method and Abrasion Test on Waterborne Sand Impact

The underwater method entails using circulating water (1200 rpm) that in turn moves the steel balls on the concrete surface, producing the desired abrasion effects. Figure 3 depicts a photo of the concrete specimen at the end of 72 h of testing. This photo indicates that the surface mortar layer has been worn away, the aggregate has been exposed, and the low strength mortar region has formed an indentation. The surface of concrete prepared with a high w/cm ratio of 0.50 exhibited a strongly nonregular structure, whereas concrete prepared with a low w/cm ratio of 0.28 exhibited a slightly nonregular surface. However, these nonregular surfaces were smooth.

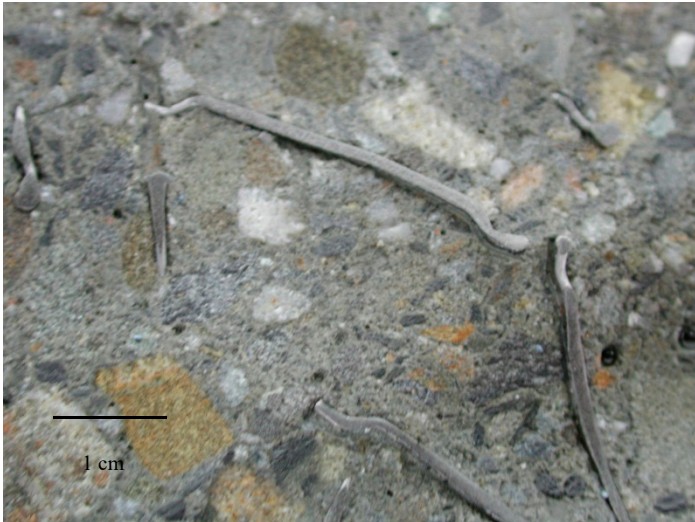

**Figure 3.** Concrete surface at the end of 72 h of testing (w/cm = 0.50, ASTM C1138).

As illustrated in Figure 4, steel fibers added to the concrete, when exposed, were flattened by the rotating steel balls and covered by the mortar during the test period. The hard and the highly erosion-resistant steel fibers acted as shields, preventing the concrete material behind the fibers from being abraded by the rolling steel balls.

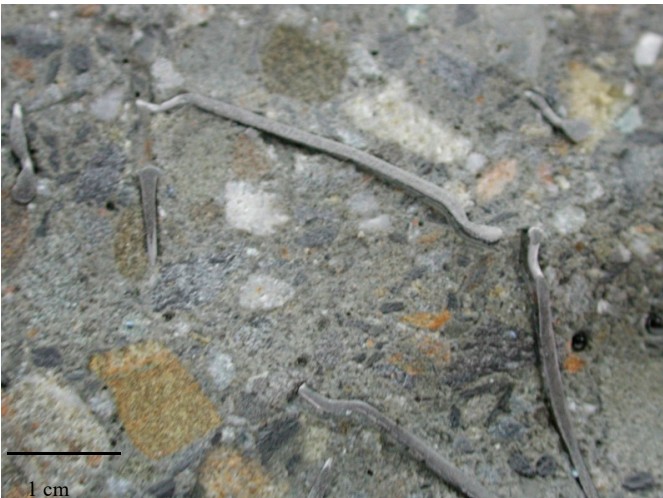

**Figure 4.** Typical abrasion site in steel fiber concrete abraded by friction abrasive action (w/cm = 0.36, ASTM C1138).

According to the aforementioned observations, the friction wear involved mainly abrasive action resulting from the rolling water and steel balls. The principal resistances were shear strength and surface hardness. Therefore, concrete made with high compressive strength or containing high hardness materials, such as chert, quartzite and steel fiber, could exhibit improved resistance to friction abrasion. Similar findings have been reported in references [5,16].

Observations of the specimen after being subjected to a waterborne sand jet test revealed that transient hydraulic rim-pulls impinged on the specimen and caused local tensile stresses in the top layer of the exposed concrete. In accordance with the theory of energy conservation, the intensity of the tensile stresses varied with respect to the impact momentum of the hydraulic jet forces. These tensile stresses are the prime culprits for causing cracks in the hardened mortar and fractures around the aggregate particles that eventually led to impact abrasion.

Figure 5 shows various impact abrasions of the concrete after testing. The cement matrix exhibited significant indenting by the exposed erodent, and the aggregate grain appeared to peel away; specifically, the mortar layer was worn away and interfacial cracks became visible on the concrete prepared with a high w/cm ratio and impacted at 90° (Figure 5a), whereas it appeared to be rather smooth on the concrete prepared with a low w/cm ratio and impacted at 45° (Figure 5b). SEM revealed the formation of cracks in the cement matrix and the interface between the aggregate grain and the cement matrix (Figure 6a) for the concrete impacted at 90°. Additionally, the concrete impacted at 90° displayed a rougher and more rugged surface than did the concrete impacted at 45° (Figure 6b).

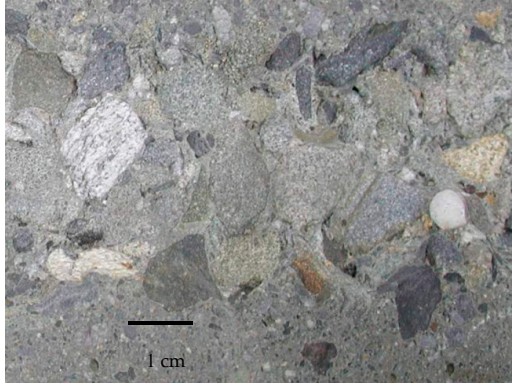

(**a**) w/cm = 0.50; impact angle = 90°

**Figure 5.** *Cont.*

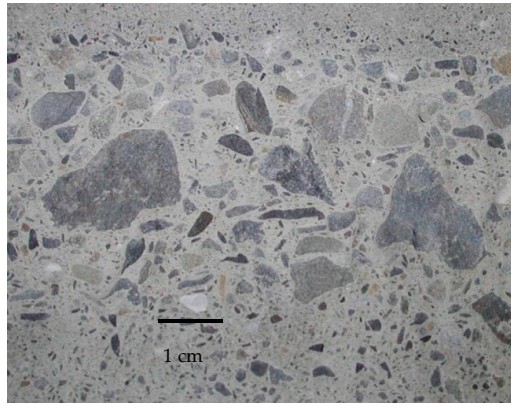

(**b**) w/cm = 0.28; impact angle = 45°

**Figure 5.** Images of worn concrete surfaces after impact test.

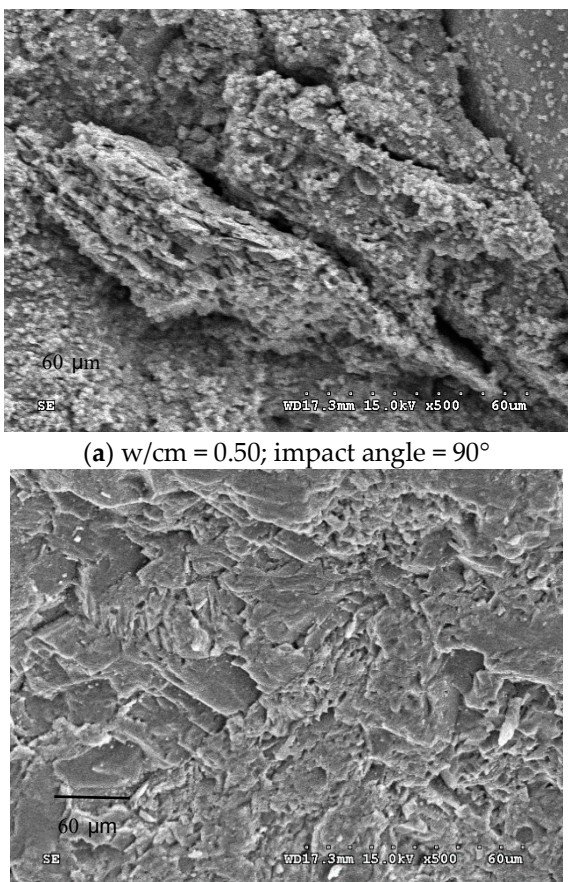

(**a**) w/cm = 0.50; impact angle = 90°

(**b**) w/cm = 0.28; impact angle = 45°

**Figure 6.** SEM images of worn concrete surface after impact test; scale: 60 μm.

The presence of steel fibers in the concrete caused some notable phenomena. As Figure 7a illustrates, the partial fiber length separated from the matrix was exposed and the front region tapered to a fine point. Figure 7b shows that the entire circumference of the fiber was exposed to a certain depth, and the amount of exposed fiber was comparatively high for the fiber concrete impacted at 90°. The exposed steel fiber could shield the concrete material from being removed. Another notable observation was the loss of bonding between both the fiber and the matrix for the concrete impacted at 45° and 90° (Figure 7c). This observation demonstrates the weak interface between the fiber and the concrete matrix.

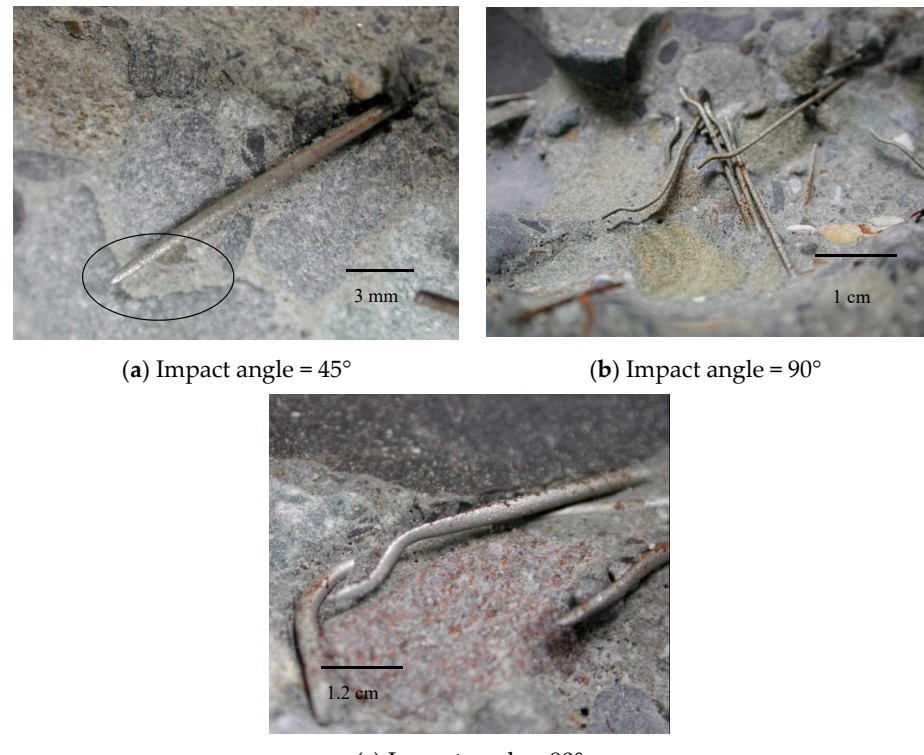

(**a**) Impact angle = 45°  (**b**) Impact angle = 90°

(**c**) Impact angle = 90°

**Figure 7.** Typical abrasion site in steel fiber concrete abraded by impact abrasive action (w/cm = 0.36, waterborne sand flow).

Overall, for the concrete impacted at 90°, the abrasive action mainly included pre-abrasion peeling engendered by water molecules and its associated hydraulic pressure, the solid particle impact, edge effects, and prising. The principal resistances were tensile strength and the matrix/aggregate bond. For the concrete impacted at 45°, the aforementioned resistances applied, as well as the friction wear engendered by waterborne sand flow and principal resistances of shear strength and surface hardness.

### 3.2. Abrasion Resistance of Concrete by Underwater Method

The abrasion resistance of concrete mixtures was determined at 28 days. It was measured in terms of abrasion loss, and the measurement revealed that the abrasion loss increased with abrasion time in all concrete mixtures. The effects of the w/cm ratio on the abrasion resistance of concrete are illustrated in Figure 8, which presents a plot of the average abrasion loss of the concrete mixtures at the 72 h test time. Reducing the w/cm ratio from 0.50 to 0.28 resulted in an approximately 50% improvement in abrasion resistance at 72 h. Furthermore, the FAR during the first 12 h was generally greater than that during the rest of the test period; this is because the surface mortar layer was easily abraded, and the FAR decreased as the mortar layer wore away, consequently resulting in the exposure of the more homogeneous specimen core. This behavior was more apparent in concrete prepared with a high w/cm ratio. Specifically, for the concrete prepared with a w/cm ratio of 0.50, the abrasion rate stabilized after approximately 48 h of the test. Concerning the concrete prepared with w/cm ratios of 0.36 and 0.28, the variations in abrasion rate became small after 24 h of the test (Figure 9).

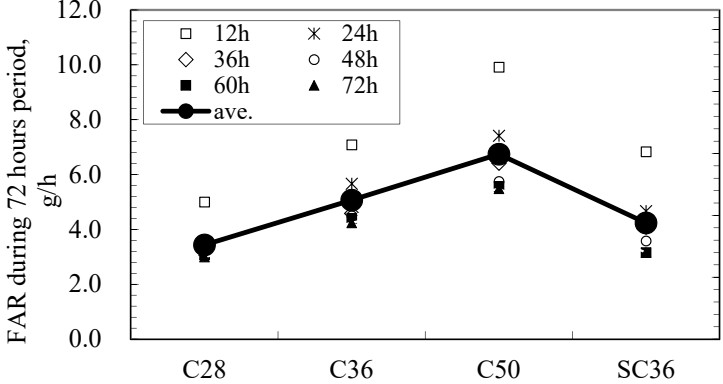

**Figure 8.** Friction abrasion rate (FAR) of concrete at different time during the 72 h period test.

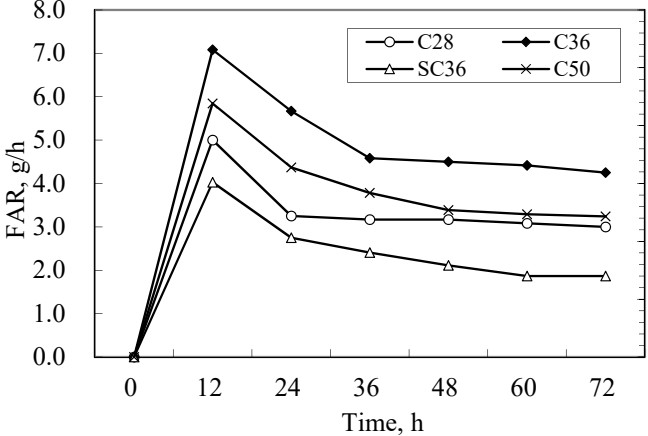

**Figure 9.** Friction abrasion rate (FAR) of concrete (SC36 with steel fibers, others without).

A comparison of the abrasion resistance of concrete with and without steel fiber reinforcement and prepared with the same w/cm ratio of 0.36 is shown in Figures 9 and 10. The concrete containing steel fiber reinforcement exhibited similar abrasion loss during the first 12 h to plain concrete. However, the abrasion rate decreased after 24 h, and the total abrasion loss of steel the fiber-reinforced concrete at 72 h was approximately 17% less than that of the plain concrete. Because of the protection provided by the hard and highly erosion-resistant steel fibers (Figure 4), the concrete material behind the fibers could not be abraded, thus improving the abrasion resistance of the concrete.

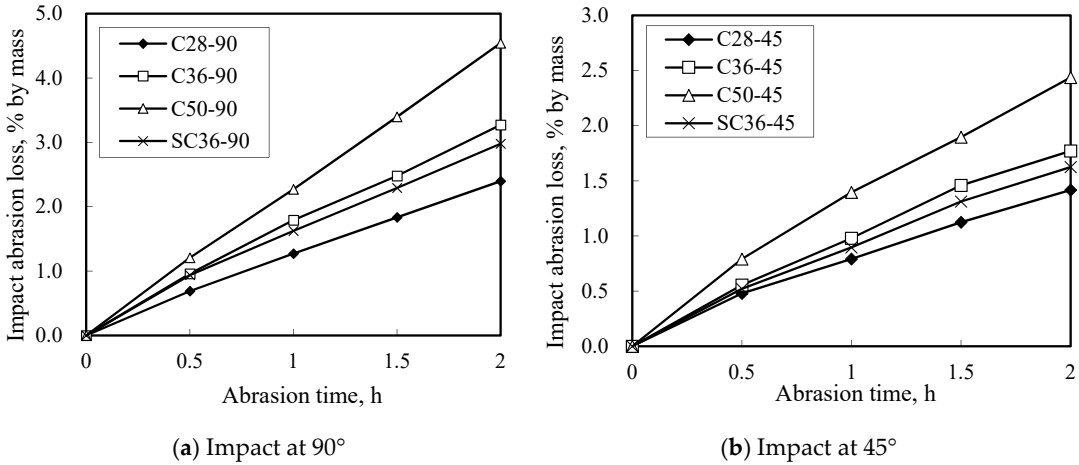

(**a**) Impact at 90°　　　　　　　　　　　　　(**b**) Impact at 45°

**Figure 10.** Impact abrasion loss of concrete.

*3.3. Impact Abrasion Resistance of Concrete by Waterborne Sand Flow*

Figure 10 shows the relationship between the w/cm ratio and impact abrasion loss. For the concrete impacted at 45°, when the w/cm ratio increased from 0.28 to 0.36 and then to 0.50, the total impact abrasion loss at 2 h increased by approximately 25% and 72%, respectively. For the concrete impacted at 90°, increasing the w/cm ratio from 0.28 to 0.36 and then to 0.50 resulted in wear resistance improvements of nearly 26% and 88%, respectively. These results reveal that concrete of low strength can be worn easily by water jets and can subsequently develop additional porosity, constituting an undesirable cycling effect. By contrast, preparing concrete with a low w/cm by adding silica fume as micro-filler and pozzolanic material can substantially reduce the overall porosity and pore sizes of the concrete, in addition to strengthening the bond between particles of the hydrated matrix [14]. Concrete prepared with a low w/cm ratio performs more favorably in resisting impact abrasions. Additionally, the impact abrasion loss was influenced by the impact angle in this study. At the end of 2 h of testing and increasing the w/cm ratio from 0.28 to 0.36 and then to 0.50, the impact abrasion loss of the concrete impacted at 90° was nearly 69%, 85%, and 86% higher than that of the concrete impacted at 45°, respectively (Figure 11). This may explain why the hydraulic pressure and its associated particle prising action on the concrete impacted at 90° were higher than those of the concrete impacted at 45°, thus increasing the impact abrasion loss. Moreover, Figure 11 shows an improvement in resistance to the impact abrasion for the steel fiber-reinforced concrete relative to that of the reference mixture with the same w/cm ratio. For the fiber-reinforced concrete impacted at 45° and 90°, the impact abrasion loss at 2 h increased by nearly 8% and 9%, respectively, relative to that of the reference mixture (C36). This may be because the addition of steel fibers and silica fume resulted in increased tensile and bending strength.

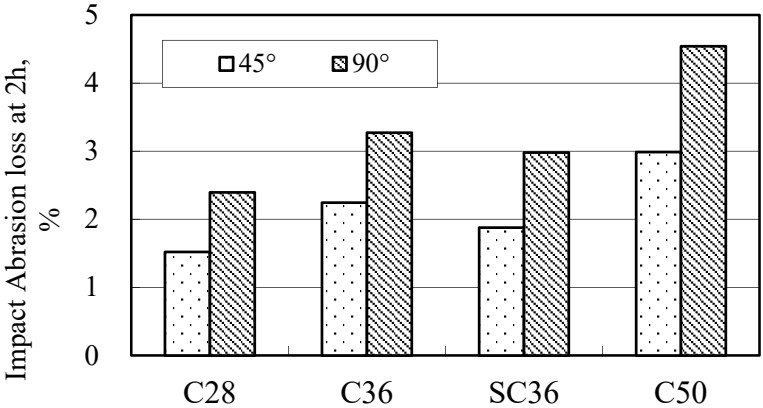

**Figure 11.** Total impact abrasion loss of concrete at 2 h.

During the impact abrasion test, concrete abrasive wear occurred readily in the small holes and pits. The IAR was also influenced by the aggregate, regardless of whether it was peeled off. When the aggregate was removed through prising, the IAR increased. Consequently, the IAR trend was nonregular (Figure 12). For the concrete made with a low w/cm ratio (0.28) and impacted at 45° and 90°, the IAR stabilized after 1 h of the test. This can be attributed to the fact that a low w/cm ratio results in a denser and stronger matrix, improving the homogeneity of concrete and thus leading to a stable abrasion rate. Regarding the other types of concrete, during the impact abrasion test, removing the aggregate engendered an increase in abrasion loss and consequently increased the IAR.

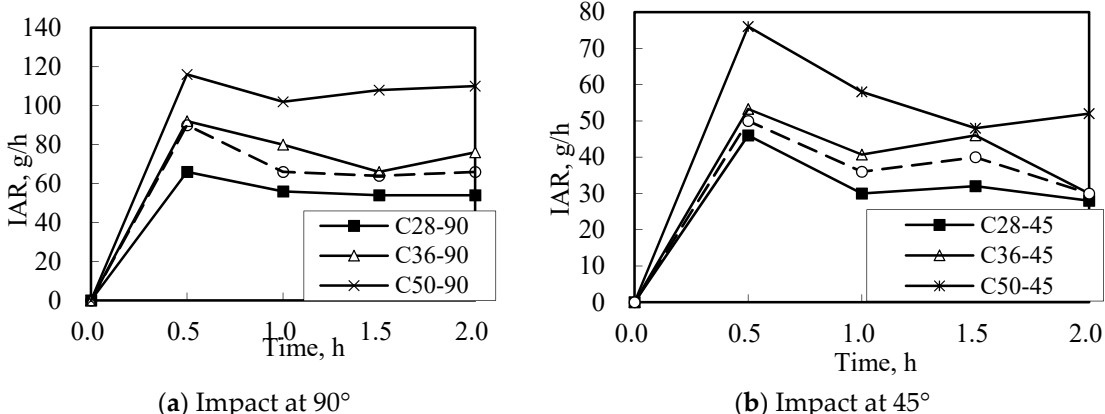

(**a**) Impact at 90°    (**b**) Impact at 45°

**Figure 12.** Impact abrasion rate (IAR) of concrete.

### 3.4. Comparing FAR with IAR

The two abrasion test methods of this investigation differed in abrasive action, and this had a significant influence on the derived average abrasion rate. In Figure 13, when the w/cm ratio increased from 0.28 to 0.36 and then to 0.50, the average IARs of the concrete impacted at 45° were approximately 9.9, 8.4, and 8.6 times the average FARs, respectively. For the concrete impacted at 90°, the gains in the average IAR were about 16.7, 15.4, and 16.1 times those of the average FAR. These results reveal that the concrete attacked by waterborne sand flow impact abrasion exhibited greater abrasion damage than did that attacked by waterborne particle friction abrasion at the same abrasion time. This result confirms that the waterborne sand more easily wears a specific region of a hydraulic structure than water flowing over a structural surface does.

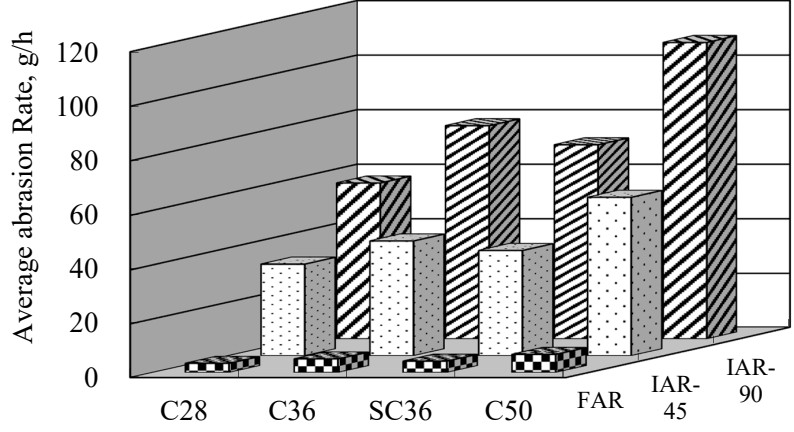

**Figure 13.** FAR and IAR of concrete.

### 3.5. Correlations between Abrasion Rate and Compressive Strength

The abrasion resistance of concrete without steel fibers with a compressive strength ranging from approximately 31 to 91 MPa was investigated. The relationship between the average abrasion rate and compressive strength of concrete is illustrated in Figure 14. The results indicated that the average abrasion rate of concrete increased as the compressive strength decreased. As the compressive strength decreased from 50 to 31 MPa, the average FAR, IAR-45°, and IAR-90° increased to 0.08, 0.80, and 1.53 per MPa, respectively, and they increased to only 0.04, 0.21, and 0.52 per MPa, respectively, as the compressive strength decreased from 91 to 50 MPa. The result confirms that compressive strength was a crucial factor affecting the abrasion resistance of the concrete without steel fibers. This can be attributed to the fact that increases in the compressive strength of concrete, which consequently

enhanced mortar strength, were comparable to those of the coarse aggregate; both the aggregate and concrete were concurrently subjected to the same abrasive forces in the water-jet test or particle wear test. Additionally, the interfacial bond strengthened, and the equivalent peeling and grinding on the aggregate and mortar enhanced the overall performance of the concrete in terms of abrasion resistance. This finding confirms the results from other studies [5,7,16].

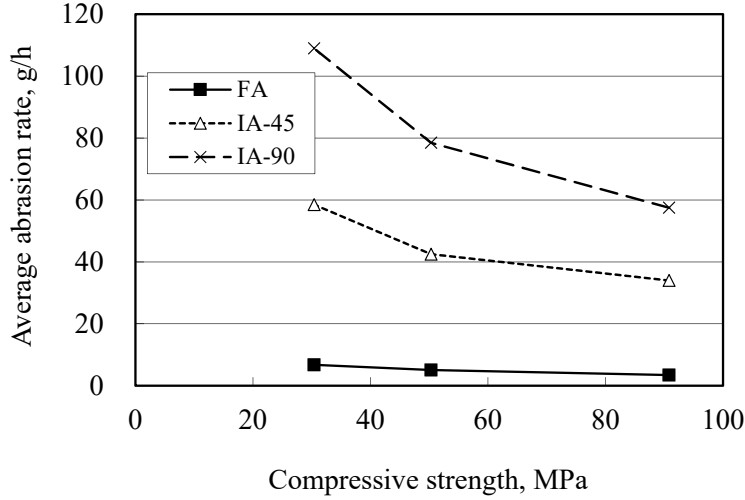

**Figure 14.** Relationship between average abrasion rate and compressive strength of concrete without steel fiber.

## 4. Conclusions

The following conclusions were formulated on the basis of the test results:

1. Both the ASTM C1138 test and the waterborne sand flow impact test were successful in evaluating the relative performance in abrasion resistance of the concrete of hydraulic structures sited on a high-sand percentage river.

2. For the ASTM C1138 underwater method, friction wear involved mainly abrasive action from the rolling water and steel balls. The principal resistances were shear strength and surface hardness.

3. For the concrete impacted by waterborne sand flow at 90°, the abrasive action mainly involved pre-abrasion peeling by water molecules and its associated hydraulic pressure, solid particle impact, edge effect, and prising. The principal resistances are tensile strength and the matrix/aggregate bond. For the concrete impacted at 45°, the aforementioned resistances applied, in addition to abrasion wear by waterborne sand flow and the principal resistances of shear strength and surface hardness.

4. Concrete attacked by waterborne sand flow impact abrasion exhibited greater abrasion damage than that subjected to waterborne particle friction abrasion with the same abrasion time. The average IARs of the concrete were approximately 8–17 times the average FARs.

5. In both the underwater and waterborne sand flow methods, the abrasion resistance of concrete without steel fibers increased as the w/cm ratio decreased, and the compressive strength of the concrete also increased. As the compressive strength decreased from 50 to 31 MPa, the average FAR, IAR-45°, and IAR-90° increased to 0.08, 0.80, and 1.53 per MPa, respectively, and they increased to only 0.04, 0.21, and 0.52 per MPa, respectively, as the compressive strength decreased from 91 to 50 MPa.

6. Concrete containing steel fibers and silica fume exhibited improved abrasion resistance. For concrete subjected to friction abrasion and impacted at 45° and 90°, the improvements in friction and impact abrasion loss were nearly 17%, 8%, and 9%, respectively, relative to those of the reference mixture (C36).

**Author Contributions:** Conceptualization, Y.-W.L.; methodology, Y.-W.L.; formal analysis, Y.-Y.L.; investigation, Y.-W.L.; resources, Y.-Y.L.; data curation, S.-W.C.; writing—original draft preparation, S.-W.C.; writing—review and editing, Y.-W.L.; project administration, Y.-W.L. All authors have read and agree to the published version of the manuscript.

**Funding:** This research was funded by National Science Council of Taiwan, grant number NSC-96-2221-E-415-004.

**Acknowledgments:** The generous donation of materials from Jack & Vicky Co., Ltd., China Hi-Ment Corporation, and Hi Con Chemical Admixture Taiwan, Ltd.

**Conflicts of Interest:** The authors declare no conflict of interest.

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
