# Peer review of "Abrasion Behavior of Steel-Fiber-Reinforced Concrete in Hydraulic Structures"

_applsci, doi:10.3390/app10165562_

Round 1
Reviewer 1 Report
It was a pleasure to read and review this article. The goal of this manuscript is attempting to experimentally investigate abrasion resistance of steel-fiber-reinforced concrete using friction abrasion and impact abrasion tests. This experimental study is interesting and the subject fits within the scope of the journal. However, the reviewer would rather hesitate to recommend the acceptance of the manuscript with the current form to the journal of Applied Sciences. The authors are recommended to consider the following suggestions for the reconsideration of the article.
- Line 82: a proper acronym of ground-granulated blast furnace slag should be GGBFS.
- Line 84: Please provide details of the used steel fiber including shape and coating materials if any.
- The chemical compositions of cement, GGBFS, silica fume should be provided.
- Table 1: 0.75Φ is not a length.
- Line 89~: silica fume is generally classified as cementitious materials. Therefore, w/cm ratio should consider the amount of silica fume.
- Line 90: Please explain why the authors replaced cement with 20% of GGBFS.
- Line 93: Please specify the requirements the authors mentioned.
- Table 2: It is not clear what (in) stands for the slump.
- Mix design: It is strange that there is no GGBFS in the mix design but in the text. It also seems odd that why the authors didn’t use silica fume in the mixture of SC36.
- Figure 2: The right-handed schematic is not clear. For example, why two mixers should be placed underneath the specimen. In addition, a caption should be used to tell the reader exactly what the schematic stands for.
- Line 151: It is not clear what is “quartz tic river sand”
- Line 152: What is ρp?
- Line 163-164: Why the range among the test results should be no greater than 45% of their average.
- Line 176: It is difficult to identify compressed parts in Fig. 4.
- The caption of Figure 4: ASTM C 1138 should be removed.
- Figure 5: A scale bar is required.
- Figure 6: The caption should clearly summarize the figures. For example, ‘after impact test’ is needed to add.
- Figure 7: The caption should clearly summarize the figures. For example, ‘after impact test’ is needed to add. And please specify the series or specimen name for the used images.
- Figure 9: Please double check the caption of y-axis. The data includes not only 72h but also 12h, 24, and so on.
- Lines 269 and 338: The authors didn’t define the reference mixture. Moreover, it seems not adequate to call it a reference mixture because C36 contains silica fume, but SC36 does not contain silica fume.
- Section 3.5: It is required to clarify that the authors were tried to investigate the effect of compressive strength of concrete without steel fiber.
- An in-depth literature review is required in the introduction. For example, recently interesting abrasion research was conducted for ultra high performance fiber reinforced cementitious composites (UHP-FRC) (Pyo et al. 2018, “Abrasion resistance of ultra high performance concrete incorporating coarser aggregate,” Construction and Building Materials, Vol. 165, pp. 11-16.)
- Minor issues: line 163 (FAR should be IAR).
Reviewer 2 Report
In the opinion of this reviewer, the following considerations are made:
Throughout the article, test tubes with different water/cement ratios are used, however in the abstract, no mention is made about the use of different water/cement ratios.
A set of test pieces with different water/cement ratios are used to carry out the tests. The ratio 0.28 is considered to be a little used ratio for structural concrete. Nor does the article give a reason for choosing the 0.36 ratio for steel fibre reinforced concrete, since other dosages are also dealt with in the same study.
The study states that sets of six specimens are made for the various tests proposed, but the results are announced on a set of three specimens, with no indication of the reason for this difference.
The description in Figures 3, 4 should specify which type of specimen (w/cm) the images correspond to, and the descriptions in Figures 5 and 8 also does not specify whether the image corresponds to one of the specimens tested or a general image.
In figure 9, the results of the friction abrasion rate (FAR) of concrete with and without stell fibers are presented, however, it is not shown in the graph if it corresponds to concrete with fibers or without fibers.
Finally, the references used are not very extensive.
Round 2
Reviewer 1 Report
The reviewer would express gratitude for preparing detailed responses from the authors. However, the concerns raised by the reviewer were still unresolved.
1) Scale bars should be added in Figure 3, Figure 4 (originally Figure 5) and Figure 7 (originally Figure 8).
2) Figure 8 (originally Figure 9): Please double check the caption of y-axis. The data includes not only 72h but also 12h, 24, and so on.
